# Proteomic Investigation of Immune Checkpoints and Some of Their Inhibitors

**DOI:** 10.3390/ijms25179276

**Published:** 2024-08-27

**Authors:** Marco Agostini, Pietro Traldi, Mahmoud Hamdan

**Affiliations:** Istituto di Ricerca Pediatrica Città della Speranza, Corso Stati Uniti 4, 35100 Padova, Italy; m.agostini@unipd.it (M.A.); mhglaxo@gmail.com (M.H.)

**Keywords:** proteomic investigations, immune checkpoints, immune system, immune therapy checkpoint inhibitors/therapies, post-translational modifications (PTMs)

## Abstract

Immune checkpoints are crucial molecules for the maintenance of antitumor immune responses. The activation or inhibition of these molecules is dependent on the interactions between receptors and ligands; such interactions can provide inhibitory or stimulatory signals to the various components of the immune system. Over the last 10 years, the inhibition of immune checkpoints, such as cytotoxic T lymphocyte antigen-4, programmed cell death-1, and programmed cell death ligand-1, has taken a leading role in immune therapy. This relatively recent therapy regime is based on the use of checkpoint inhibitors, which enhance the immune response towards various forms of cancer. For a subset of patients with specific forms of cancer, these inhibitors can induce a durable response to therapy; however, the medium response rate to such therapy remains relatively poor. Recent research activities have demonstrated that the disease response to this highly promising therapy resembles the response of many forms of cancer to chemotherapy, where an encouraging initial response is followed by acquired resistance to treatment and progress of the disease. That said, these inhibitors are now used as single agents or in combination with chemotherapies as first or second lines of treatment for about 50 types of cancer. The prevailing opinion regarding immune therapy suggests that for this approach of therapy to deliver on its promise, a number of challenges have to be circumvented. These challenges include understanding the resistance mechanisms to immune checkpoint blockade, the identification of more efficient inhibitors, extending their therapeutic benefits to a wider audience of cancer patients, better management of immune-related adverse side effects, and, more urgently the identification of biomarkers, which would help treating oncologists in the identification of patients who are likely to respond positively to the immune therapies and, last but not least, the prices of therapy which can be afforded by the highest number of patients. Numerous studies have demonstrated that understanding the interaction between these checkpoints and the immune system is essential for the development of efficient checkpoint inhibitors and improved immune therapies. In the present text, we discuss some of these checkpoints, their inhibitors, and some works in which mass spectrometry-based proteomic analyses were applied.

## 1. Introduction

Frequent genetic and epigenetic alterations that are intrinsic to most cancer cells produce a high number of tumor-associated antigens/proteins that the host’s immune system can recognize. Such recognition induces tumors to develop specific immune resistance mechanisms. The introduction of immune checkpoints therapy (ICT) that regulate immune responses gave fresh hope to many cancer patients, particularly those who suffer from metastatic conditions. This highly promising development started over a decade ago when, in 2011, the Food and Drug Administration (FDA) approved ipilimumab as the first antibody capable of blocking the immune checkpoint CTLA-4 (cytotoxic T lymphocyte antigen 4). This approval was rapidly followed by the development of monoclonal antibodies targeting PD-1 (programmed cell death-1), PD-L1 (programmed cell death ligand-1), CTLA-4, and CD47 [1,2,3,4]. These developments were justly considered a new milestone in the treatment of advanced metastatic solid tumors. One of the early successes of ICT was reported for metastatic skin melanoma, where the 5-year survival rates for this tumor reached an unprecedented 52% when applying the combined CTLA-4 and PD-1 blockade approach [5]. Unfortunately, this early success was not repeated in many patients with advanced metastasis conditions, who either did not respond well or simply did not respond at all (refractory patients) [6]. Unlike chemotherapy, which directly attacks tumor cells’ growth and survival, immunotherapies target the tumor indirectly by boosting the antitumor immune responses that spontaneously arise in many patients. Currently, there is sufficient evidence to suggest that cancerous cells are genetically unstable, which influence their uncontrolled proliferation and the expression of antigens that can be recognized by the immune system. These antigens include normal proteins overexpressed by cancer cells and novel proteins that are generated by mutation and gene rearrangement [7]. This and other similar works enforce the observation that various proteins and antigens will have a relevant role in the dynamic interplay between cancer cells and the immune system during the course of the disease.

Mass spectrometry (MS) is a central component in present-day proteomic activities. These activities can furnish crucial information on proteins, antigens, and interacting ligands, which influence the relationship (interaction) between cancer cells and the immune system. Quantitative assessment of these biomolecules and their likely post-translational modifications (PTMs) are key elements in attempts to understand the mechanism(s) of resistance to the inhibitors of immune checkpoints. For over three decades, various research activities have demonstrated that MS-based proteomics is a powerful tool for the analyses of complex protein mixtures, both artificial as well as those found in complex biological samples. These analyses can provide information related to the proteins’ expression levels, the associated PTMs, protein–protein and protein–ligand interactions, protein assemblies, protein degradation, and proteins’ conformational dynamics. The current literature indicates that these capabilities have not yet been fully applied to challenges associated with the checkpoint inhibitors and the antigens recognized by the immune system of cancer patients. This observation is rather surprising for two reasons. First, tumor antigens and immune checkpoints molecules are peptides and proteins, respectively, and, therefore, MS-based proteomics is expected to play a major role in their characterization. Second, most immune checkpoints experience various PTMs, including extensive glycosylation, phosphorylation, and acetylation. Mass spectrometry is recognized as the method of choice for the identification and localization of both known and unknown protein PTMs. The current literature shows that the immunosuppression activity of PD-L1 is strongly modulated by a number of PTMs, including ubiquitination and *N*-glycosylation. More detailed descriptions of the role of these modifications on the immune activities of PD-L1 and PD-1 are given in Section 2.2.

The main analytical platform for the investigation of both enzymatically digested as well as intact proteins within a mixture was introduced almost three decades ago under the name of the “shotgun” approach [8]. In this approach, liquid chromatography (LC) coupled with tandem mass spectrometry (MS/MS) is used to investigate complex protein mixtures. The LC phase separates a mixture of peptides generated by enzymatic digestion of a protein mixture. In the following step, the separated peptides are injected into an electrospray ion source to generate gas-phase multiple charged ions. This mixture of charged peptides is then separated according to their mass to charge (*m*/*z*) ratios and focused into a collision cell to be fragmented and subsequently detected and interpreted. In modern mass spectrometry, the capability for separation prior to fragmentation is enhanced by adding an ion mobility component, which separates the gas-phase macromolecular ions according to their mass, charge, size, and shape [9]. To perform MS/MS, modern mass spectrometers use various modes of ion activation/fragmentation, including collision gas to perform collision-induced dissociation (CID) both at low and high levels of collision energy; electron-based fragmentation, which can be performed through electron transfer (ET) [10] or electron capture (EC) [11]; and UV photodissociation (UVP) [12]. Under the general term “shotgun”, there are three different methods of analysis: “top-down”, “middle-down”, and” bottom-up” [13,14].

## 2. Discussion

Over the last 10 years, there have been two important developments, which have impacted current efforts to improve immune therapy in particular and to advance personalized cancer treatment in general. The first development was the shift from tissue biopsy to liquid biopsy as a method of sampling cancer. The latter approach offers a number of advantages compared with tissue biopsy. This method facilitates non-invasive repetitive sampling throughout the course of the disease. This method also provides continuous updates on the evolution of the tumor, giving a robust indication on the response to therapy and an early warning of emerging drug resistance. Analysis of the genetic and proteomic entities circulating in liquid biopsy samples furnish detailed information on the heterogeneity of the tumor, which are necessary for correct stratification of the patients, monitoring the response to therapy, and early identification of emerging drug resistance. Among the technologies applied in the analyses of these biopsies are next-generation sequencing (NGS), also known as massively parallel sequencing [15] and MS-based proteomics [16].

The second relevant development was the approval by the Food and Drug Administration (FDA) of ipilimumab as the first antibody capable of blocking the immune checkpoint CTLA-4 (cytotoxic T lymphocyte antigen 4) [17]. It is interesting to note that the combination of the two developments was recently used in the prognosis of patients with metastatic non-small cell lung cancer (NSCLC) [18]. The authors used liquid biopsies to examine the correlation between circulating tumor cells (CTCs) and concentrations of PD-1 with overall survival (OS) and progression-free survival (PFS).

### 2.1. Predictive Biomarkers in Response to Immune Checkpoint Inhibitors

The binding of T cells’ immune checkpoint proteins to their ligands is known to facilitate immune evasion by tumor cells. To block this evasion, a number of immune checkpoint inhibitors have been developed. The United States Food and Drug Administration (FDA) has approved several immune checkpoint inhibitors (ICIs) for cancer therapy. Three of these inhibitors target PD-1 (nivolumab, pembrolizumab, and cemiplimab), three other inhibitors target PD-L1 (avelumab, atezolizumab, and durvalumab), and ipilimumab targets CTLA-4 [19,20]. More recently, the FDA also approved relatlimab, an immunotherapy that targets the lymphocyte activation gene 3 (LAG-3) immune checkpoint pathway. LAG-3 is widely recognized as a potent inhibitory receptor that is highly expressed by exhausted T cells [21,22]. A combination of anti-LAG-3 and anti-PD-1 is approved for the treatment of unresectable or metastatic melanoma in individuals aged 12 and older. 

Recent literature on immune checkpoint inhibitors has highlighted a number of challenges: (i) understanding the mechanism(s) of resistance to the blockade of immune checkpoints; (ii) identification of reliable biomarkers for tumors’ response to inhibitors and their effect on the evolution of the disease; (iii) more specific inhibitors, which can benefit a wider range of cancer patients with minimal adverse side effects; and (iv) early identification of the patients likely to respond to ICI therapy.

The identification of reliable predictive biomarkers of anti-PD-1/PD-L1 remains one of the priorities of immunotherapy. This is because the existing inhibitors approved by the FDA are costly, are associated with potentially severe side effects, and only benefit a small subset of patients. It is therefore imperative to identify biomarkers that discriminate between responders and non-responders. The current literature reports a number of genomic, proteomic, and transcriptomic investigations targeting various aspects of these challenges. Some of these works are discussed below. To underline the contribution of MS-based proteomics, a number of analyses are considered below.

In a recent study, samples were collected from 23 patients with non-small cell lung cancer (NSCLC) [23]. These patients were treated with anti-PD-1/PD-L1 monotherapy and followed up for 3 years. The proteomic profile of the tumor was examined by mass spectrometry (MS). These data, together with previously generated RNA sequencing data for 27 patients treated with anti-PD-1/PD-L1 therapy, were combined to establish what the authors called an integrated gene network. The authors reported the implementation of weighted correlation network analysis (WGCNA) and an elastic network to screen the top gene modules for predicting treatment-responsive patients. The main conclusions of this study were that the proteomic gene expression profile could better predict the durable response of NSCLC patients to anti-programmed cell death-1 (anti-PD-1)/programmed cell death ligand 1 (anti-PD-L1) therapy. The same study suggested that relying on PD-L1 alone as a biomarker to assess the response of NSCLC patients to this form of immunotherapy may not be sufficient due to the complex microenvironment of this form of cancer. Considering these deductions, the following observations can be made. First, the main feature of this study is the combined use of proteomic, genomic, and transcriptomic technologies in the analysis of samples from cancer patients. The complexity of cancer and its microenvironment necessitate the use of different technologies to obtain reliable answers to a wide range of problems. Second, as suggested by the authors of the study, future proteomic analyses with larger sample sizes should be conducted to further confirm the clinical application of proteomic gene panels in predicting the efficacy of immunotherapy among NSCLC patients. Third, the authors suggested that the failure of PD-L1 as a single biomarker for the efficacy of immunotherapy for NSCLC patients is linked to a complex microenvironment of this form of cancer. This observation is legitimate, as the role of tumor microenvironment in drug resistance is well documented; however, it is reasonable to add that the tumor microenvironment is one of a number of reasons for the failure of a single biomarker.

In another recent study, longitudinal plasma analysis was used to identify biomarkers and combinational targets for anti-PD1-resistant cancer patients [24]. The authors examined 339 plasma samples derived from 193 patients suffering from NSCLC, alveolar soft part sarcoma (ASPS), and lymphoma. Plasma samples were collected from these patients before and after anti-PD-1 therapy. Proteins within the investigated samples were detected using data-independent acquisition-mass spectrometry and antibody microarrays. These analyses generated the proteomic profiles of responders and non-responders covering an abundance of 10–12 orders of magnitude. By comparing the differences in the expression of the plasma proteome between patients who responded to the treatment and those who did not, the authors concluded that the *Th17*, *IL-17*, and *JAK*-*STAT* signal pathways were upregulated in the non-responder group, while cellular senescence and transcriptional mis-regulation pathways were activated in the responders. Before commenting on this interesting study, the following observation is useful to cons+ider: previous genomic and transcriptomic studies have identified biomarkers for predicting the response to anti-PD1 therapy. To date, three biomarkers have been approved by the United States Food and Drug Administration (FDA) [25], namely PD-L1 expression on tumor cells, microsatellite Instability/defective mismatch repair (MSI/dMMR) [26] and tumor mutational burden (TMB) [27]. This proteomic study [24] is a clear example of how MS-based proteomics can complement genomic and transcriptomic searches for predictive biomarkers. The study, however, had advantages and disadvantages. The detection of proteins in plasma samples was non-invasive and simpler compared with tissue samples, which would render the approach easier to adapt in clinical studies. That said, the detection of low-abundance proteins in biological samples is highly challenging. To mitigate this drawback, the authors developed an in-depth serum proteomic platform using data-independent acquisition mass spectrometry (DIA-MS) and customizable antibody microarrays that could detect serum proteins covering 10–12 orders of magnitude.

Serum amyloid P component (SAP), also known as pentraxin-2, is a member of the pentraxin protein family with an established relationship with the immune response. This protein was described as a potential biomarker for a reduced response to PD-1–based interventions [28]. The authors used affinity purification–mass spectrometry to investigate proteins associated with PD-1. This study revealed that the adaptor protein SAP inhibits PD-1’s functions by blocking the enzymatic interaction between the phosphatase SHP2 and a subset of its substrates that also bind to SAP. The same study reported that SAP contrasts with PD-1’s function by acting as a molecular shield of the key tyrosine residues that are targets for the tyrosine phosphatase SHP2, which mediates PD-1’s inhibitory properties.

A meta-analysis across patients with 27 types or subtypes of tumors showed that the tumor mutational burden (TMB) is correlated with the response rate following anti-PD-1 or anti-PD-L1 therapy [29]. The same meta-study supported the observation which suggests that a high TMB increases the likelihood of neoantigens’ formation and is therefore associated with a greater CD8^+^ T cell (killer T cell) response upon checkpoint inhibition [29,30]. Another study has shown that individual mutations associated with downstream effects on the TMB can also predict the response. For example, mutations in the two genes *POLE* and *POLD1*, which encode DNA polymerase, are essential for proofreading and fidelity in DNA replication, leading to an increase in replication errors and, consequently, a greater TMB. These mutations have been found to be associated with a positive response to ICIs across different forms of cancer [31,32]. Conducting a combined analysis using a large dataset, the authors reported that *POLE*/*POLD*1 mutations were promising as potential predictive biomarkers for positive outcomes of ICI [32]. It has to be said that the predictive value of the TMB can be negatively influenced by the presence of intertumoral heterogeneity (ITH). A high level of ITH indicates that the neoantigen may only be limited to a subset of cells and, therefore, the immune response may not be effective against the whole tumor [30]. This observation is in agreement with earlier reports, which showed that the combined expression of the TMB only predict a subset of responses, but they fail to predict all responses to checkpoint and *PDL1* can blockade [33].

In a more recent meta-study, biomarkers for the response to anti-PD-1/anti-PD-L1 immune checkpoint inhibitors were discussed [34]. The authors attempted to provide a comprehensive evaluation of the current state of the predictive utility of the most common biomarkers and some emerging ones for the response to ICI treatment. The authors conducted a very large meta-analysis of predictive biomarkers for ICI therapy to date, including 100 peer-reviewed studies with data from 18,792 patients. Diseases discussed in the study included non-small cell lung cancer, melanoma, urothelial cancer, head and neck cancer, and colorectal cancer. This meta-analysis concluded that mIHC/IF, multimodal biomarkers, TMB, and PD-L1 IHC adequately captured responders and non-responders across all cancer types included. Between the two most frequently investigated biomarkers, TMB outperformed PD-L1 IHC when all cancers were combined. These two biomarkers also adequately captured responders and non-responders across NSCLC and melanoma. To underline the role of MS-based proteomics in the investigation of biomarkers for the response to anti-ICIs, a number of recent investigations are given in Table 1.

A number of recent investigations using MS-based proteomics for the investigation of biomarkers for the response to anti-ICIs are given in Table 1. The therapeutic limitations of conventional IgG-based monospecific antibodies encouraged the development of a new generation of antibodies. These bispecific antibodies (bsAbs) can bind to two different epitopes on the same or on different antigens. Through this dual specificity for soluble or cell-surface antigens, bsAbs exert activities superior to those of natural antibodies. These antibodies can be relatively small proteins, merely consisting of two linked antigen-binding fragments, to large immunoglobulin G (IgG)-like molecules with additional domains attached [39,40]. Over the last 10 years, these molecules have generated substantial therapeutic interest. For example, at the end of last year, 14 bsAbs have been approved, 11 for cancer treatment and the other three for non-oncological indications [40]. Table 2 gives a list of bsAbs which have been approved for cancer therapy as well as for non-oncological indications. Recent literature attributes the success of these emerging antibodies to their capability to mediate therapeutic effects beyond those of natural monospecific antibodies [39,41]. It is hypothesized that bsAbs can recruit immune effector cells to cancer cells or by targeting different signaling pathways from a single molecule. Currently, there are insufficient clinical data to compare the therapeutic efficacy of bsAbs with monospecific antibodies.

### 2.2. Post-Translational Modifications of PD-L1/PD-1 and Their Potential Role in Cancer Therapy

It is widely accepted that proteins’ activities are not only controlled by the rates of protein biosynthesis and protein degradation but also by other processes, particularly PTMs. These protein modifications can modulate molecular interactions, redirect cellular proteins’ localization, and promote or inhibit interactions with other proteins [42]. The central role of these modifications in diverse cellular functions explains their frequent implication in human diseases, including various forms of cancer. Existing literature has demonstrated that extracellular proteins are more exposed to various PTMs. Many extracellular as well as cell surface proteins experience extensive PTMs, including glycosylation and disulfide bonds. These modifications contribute to proteins’ structural stability and enhanced solubility, and, in certain cases, serve as recognition elements for binding with other biomolecules. Such binding enables these modified proteins to exert their functions in the extracellular environment [43,44].

Programmed cell death protein 1 (PD-1) is a membrane protein of the immunoglobulin superfamily. This cell surface protein has 288 amino acids and is mainly expressed on T and B cells [45]. This immunosuppressive protein was cloned over 30 years ago from the apoptotic mouse T cell hybridoma [46]. PD-1 consists of an IgV-like domain, a transmembrane domain, a cytosolic domain, and a stalk region that separates the IgV domain from the plasma membrane [47,48]. The calculated molecular weight is 17090.1 Da, while sodium dodecyl sulphate polyacrylamide gel electrophoresis (SDS-PAGE) and MS measurements gave molecular weights ranging from 35–45 and 18–19 kDa, respectively. The difference between the calculated and measured MWs is mainly attributed to different PTMs, in particular, N-linked and O-linked glycans [49]. Programmed death ligand1 (PD-L1) is a Type 1 transmembrane protein, has a MW of 40 kDa, and is known to play an important role in suppressing the immune system when the system is challenged by autoimmune diseases and viral infections [50] PD-L1 is mainly expressed on the cell surface of tumor cells or antigen-presenting cells; the formation of the PD-1/PD-L1 complex transmits an inhibitory signal that reduces the proliferation and activity of killer T lymphocytes [51].

Increasing evidence indicates that extracellular interactions between programmed death ligand-1 (PD-L1) and programmed cell death protein-1 (PD-1) contributes to tumors’ evasion of the immune reaction. It has been also reported that the immunosuppression activity of PD-L1 is strongly modulated by a number of PTMs, including ubiquitination and *N*-glycosylation [52] The same study showed that glycogen synthase kinase 3β (GSK3β) interacts with PD-L1 and induces phosphorylation-dependent proteasome degradation of PD-L1 by β-TrCP. The glycosylation of PD-L1 is known to influence its interaction with PD-1, which, in turn, impacts on T cell-mediated immune responses [52,53]. 

Mass spectrometry remains the method of choice for the identification and localization of both known and unknown protein PTMs. Glycosylation remains one of the modifications which impose stringent conditions of analysis both during the analyses and in the interpretation of the generated MS and MS/MS data. This is because a single glycosylation site can be occupied by multiple heterogeneous glycated structures, which inevitably result in different glycoforms of the same protein This effect causes the distribution of the glycopeptide signals and the inevitable signal reduction of the individual structures [54]. Interpretation of the collision-induced dissociation of glycopeptides is also problematic because of the preferential cleavage of glycosidic bonds in carbohydrate moieties [55]. Despite these technical difficulties, MS-based analyses remain the method of choice for the characterization of protein glycosylation and other PTMs. 

Both PD-1 (also known as CD279) and PD-L1 are known to experience a number of PTMs, including, glycosylation, phosphorylation, ubiquitination, and acetylation. Understanding the influence of these modifications on the PD-L1/PD-1 signaling pathway and its biological significance is attracting an increasing interest, particularly in the search for immunotherapies. This statement is supported by emerging studies indicating that PTMs of both molecules are important regulatory mechanisms that modulate immunosuppression in cancer patients. To emphasize this increasing interest, it is sufficient to consider research statistics published 4 years ago, showing 5000 patents, with over 2000 focusing on the application of PD-1 in clinical trials and more than 1000 related to the application of PD-L1 antibodies in clinical trials [56]. 

Post-translational modifications (PTMs), including glycosylation, phosphorylation, ubiquitination, and palmitoylation, play a significant role in regulating PD-1 proteins’ stability, localization, and interprotein interactions. Targeting the PTM of PD-1 in T cells has emerged as a potential strategy to overcome PD-1-mediated immunosuppression in cancer and enhances antitumor immunity. Some of these modifications associated with the various domains of PD-1 are presented in Figure 1. 

Although, it has been amply demonstrated that mass spectrometry is a powerful tool for analyzing the glycosylation of proteins [57,58], published works on mass spectrometry-based analyses of PD-1’s post-translational modifications remain extremely limited. One of these studies used various enzymes in combination with intact protein mass spectrometry analyses to probe the O-glycosylation of PD-1 [49]. As indicated in Figure 1, this protein consists of an IgV-like domain, a transmembrane domain, a cytosolic domain, and a stalk region that separates the IgV domain from the plasma membrane. Recent publications on regulation of the PD-L1/PD-1 signaling pathway has shown that PTMs of PD-L1 or PD-1, including glycosylation, ubiquitination, phosphorylation, and acetylation, may play an important role in regulation of the PD-L1/PD-1 signaling pathway and the antitumor activities of T cells [59]. The results reported in the same study underlined the role of mass spectrometry in the identification and localization of previously unidentified PTM. We have to bear in mind, as mentioned above, that glycosylation is one of the modifications which are highly demanding for MS-based proteomic analyses.

### 2.3. Mass Spectrometry-Based Analyses of Some Proteins Relevant to Immune Responses

Mass spectrometry (MS)-based proteomics is assuming an increasing role in the discovery of biomarkers. The breakthrough of immunotherapies came with the introduction of immune checkpoint inhibitors (ICIs), such as anti-CTLA4, anti-PD1, and anti-PDL1, which block inhibitory immune molecules, unleashing the activation of T cells [60]. The main mechanism of this therapy is based on boosting potentially tumor-reactive T cells directly in the patient’s body. To underline the potential contribution of MS-based proteomics, a number of studies are considered here.

Cytotoxic T lymphocyte-associated protein 4 (CTLA-4) is a checkpoint protein expressed on the surface of the T cells and has a strong influence on the regulation of the immune response. It is well documented that blocking this protein using existing immunotherapy can restore T cells’ activities, which boost the immune response to tumors. The expression levels of CTLA-4 in cells have been measured using various methods, including, quantitative polymerase chain reaction (qPCR) at mRNA levels, flow cytometry, Western blotting, and immunohistochemistry, including quantitative digital image analysis [61,62]. In a relatively recent study [63], the authors reported the development of a highly sensitive LC/MS assay for the quantification of CTLA-4 in human T cells. Quantitative assessment of this protein in T cells is relevant to the understanding of the pharmacodynamics, efficacy, and safety of CTLA-4-based therapies. The use of LC-MS and MS/MS offer higher sensitivity and higher specificity compared with the methods cited above. The experimental setup used by these authors is partially represented in Figure 2. The authors used this method to assess the level of CTLA-4 from various T and B cells isolated from human blood samples. These quantitative measurements of the level of CTLA-4 in human immune cells revealed a detection limit of this protein as low as 5 copies per cell. This study was the first to demonstrate the application of MS-based methods to quantify an important immunotherapeutic target in human T and B cells. The authors used a standard procedure for analysis of the proteins. The extracted proteins were dissolved directly in 8 M urea, then denatured, alkylated, digested, and examined with LC-MS/MS. This procedure minimized the loss of CTLA-4 during samples’ preparation and its extraction.

Second, the assay measured CTLA-4 protein in human T cells isolated from human whole blood samples. These samples were inherently limited in quantity, particularly those from cancer patients. To achieve higher sensitivity and low sample consumption, the authors used two-dimensional nano-LC-MS. The samples under investigation were enriched in trap cartridges prior to elution into the nano-LC column and injection into the ion source for MS analysis.

Another protein that is relevant to the understanding of immune responses is DC160. This glycoprotein is a member of the immunoglobulin superfamily and has four isoforms, which differ by the presence or absence of an immunoglobulin-like domain and the mode of anchoring in the cell membrane [64]. CD160 is considered a T cell coinhibitory molecule that interacts with the herpes virus entry mediator (HVEM) on antigen-presenting cells to provide an inhibitory signal to T cells. The structure of this protein was first determined 5 years ago [65]. This protein is considered to be a signaling molecule that interacts with HVEM [66] and contributes to a wide range of immune responses, including T cell inhibition and activation of natural killer cells. Understanding the complexation between CD160 and HVEM can furnish relevant information on certain interactions between cancer and the immune system. The interaction between CD160 and HVEM was investigated by multiple analytical techniques, enzyme-linked immunosorbent assay (ELISA), hydrogen/deuterium exchange, affinity chromatography, mass spectrometry, and molecular modeling [67]. The use of HDX-MS provided key information about the tertiary structure of CD160, predicting the 3D structure of the CD160–HVEM complex. The same analysis determined the binding between the CD160 and HVEM, which were in good agreement with the theoretical results. The 3D structure of CD160 complex with HVEM derived from HDX-MS analyses was found to be similar to the BTLA–HVEM complex based on a crystallographic analysis. This similarity merits further considerations regarding BTLA. 

The B and T lymphocyte attenuator (BTLA also known as CD272) is an important co-signaling molecule. This protein belongs to the CD28 immunoglobulin superfamily and shares structural and functional similarities with PD-1 and CTLA-4. The BTLA protein consists of a signal peptide, an IgC-like extracellular domain, a transmembrane domain, and a cytoplasmic domain (Figure 3). The BTLA protein is also produced in a soluble form (sBTLA) as a result of alternative RNA splicing. This form lacks the transmembrane region, due to the lack of Exon 3. The exact mechanism of producing this soluble form has not been published yet. This protein is considered to be a crucial checkpoint, regulating stimulatory and inhibitory signals in immune responses [68]. BTLA-targeted therapies have shown improved treatment outcomes and enhanced antitumor immunity. The same protein can be detected in most lymphocytes and induces immunosuppression by inhibiting activation and proliferation of B and T cells [69]. The monomeric structure of BTLA resembles the structure of CD160; furthermore, the crystal structure of the CD160–HVEM complex revealed a binding mode resembling the structure of the BTLA–HVEM complex. This structural similarity between monomeric and complexed structures may explain why both molecules interact with HVEM in a 1:1 stoichiometry [61]. 

HDX-MS was used to gain structural as well as conformational information on the retinoic acid inducible gene-I (RIG-I). This molecule ensures immune surveillance of viral RNAs bearing a 5′-triphosphate (5′ppp) moiety. HDX-MS analyses were used to reveal dysregulated checkpoints that result in the recognition of self-derived RNA during RIG-I-mediated autoimmunity [70]. RIG-I is a critical receptor in the induction of innate immune responses, but mutations in RIG-I can be associated with the hyperactive signaling associated with autoimmune diseases [71,72]. In a more recent study, the authors used HDX/MS and single-molecule magnetic tweezers (MT) to precisely examine how small conformational changes in the helicase insertion domain (HEL2i) promote impaired ATPase and erroneous RNA proofreading activities. The magnetic tweezer technique can provide insights into the physical and mechanical properties, as well as the conformational dynamics of single macromolecule [73].

### 2.4. Comments

#### 2.4.1. Drug Resistance to Immune Therapies

Drug resistance, both intrinsic and acquired, to existing therapies, including immune therapy, remains the main challenge to efforts to defeat cancers. Most of the recent findings in oncological research suggest that cancer’s survival and spread depend on the ability of tumor cells to avoid immune recognition. This means that a deeper understanding of cancer’s immunity and tumor immune escape mechanisms is central to the development of efficacious immunotherapeutic approaches. Most high-throughput studies over the past decade have focused on omics-based characterization at the DNA and RNA level. However, proteins are the molecular effectors of genomic information; therefore, the study of proteins provides a deeper understanding of cellular functions.

The main message emerging from extensive research efforts over the last 50 years is that drug resistance is a multifactorial phenomenon and has to be treated as such. Numerous works have demonstrated that treating drug resistance as the consequence of a single factor can only lead to partial and short-term success, which benefits a small subset of patients. Research efforts to circumvent drug resistance is a long and evolving process; therefore, it is important to learn from previous disappointments. We may find it useful to consider a concrete example, which is representative of such disappointments. 

The ATP-binding cassette (ABC) transporter family of proteins is capable of regulating the flux across the plasma membrane of structurally different chemotherapeutic agents. So far, there are 48 known members of this family [74]. P-glycoprotein (also known as ABCB1 and MDR1) was the first identified mammalian ABC transporter protein and, so far, is the most studied member of this superfamily. Furthermore, there is strong evidence that high expression of this protein remains one of the main reasons for resistance to both chemotherapy and to targeted therapy in many types of cancer. A hallmark characteristic of this transporter protein is its ability to bind and transport a wide range of structurally different molecules in the molecular mass range of 100 to 4000 Da, a range which covers most, if not all, anticancer and antimicrobial drugs currently in use [75]. The strong link between the overexpression of this protein and resistance to chemotherapy sparked intense research activities to discover and develop inhibitors of P-glycoprotein. These research efforts extended for over 40 years, and four generations of potential inhibitors of P-glycoprotein have been developed and tested [76]. During this period, many potential inhibitors were clinically tested, but none of them obtained the approval of either the FDA (Food and Drug Administration) or the EMA (European Medicines Agency). Most experts in the field argued that the failure of these inhibitors to restore sensitivity to chemotherapy reside in their poor selectivity, low potency, inherent toxicity, and/or adverse pharmacokinetic interaction with the administered therapy. The reasons behind the disappointing outcome of long years of research have been addressed elsewhere [77,78]. The opinion of the present authors is that one of the main reasons for such poor outcomes was considering a single mechanism (the overexpression of P-glycoprotein) to be responsible for drug resistance, which is known to be caused by multiple mechanisms. The scenario above can be easily compared with current activities to discover and develop specific, more efficient immune checkpoint inhibitors. It is true that we are talking about two different therapy regimes. In radiotherapy and chemotherapy, tumor cells are directly attacked, while in immunotherapy, the tumor is indirectly attacked by boosting the antitumor immune responses that spontaneously arise in cancer patients. That said, both therapy regimes are based on the inhibition of molecules identified as the main cause of disease. The statistical and clinical data accumulated so far indicate that therapy approaches based on the inhibition of a single cause of some forms of cancer can achieve some success for a subset of patients; however, for the majority of patients, particularly those in advanced stages of the disease, developing resistance to these therapies is almost inevitable.

#### 2.4.2. The Promise of Bispecific Antibodies

Recently, bispecific antibodies have emerged as a class of molecules capable of exerting activities beyond those exerted by natural monospecific antibodies. Initially developed for retargeting T cells to tumors, with the first bsAb approved in 2009 (catumaxomab, withdrawn in 2017) [41]. Catumaxomab is a trifunctional monoclonal antibody with two different antigen-binding sites and a functional Fc domain [79]. The two specific antigen-binding sites bind to epithelial tumor cells via the epithelial cell-adhesion molecule (EpCAM) and to T cells via CD3 [80]. The limited success of conventional IgG-based antibodies as inhibitors of immune checkpoints in various forms of cancer have generated increased interest in inhibitors based on different molecular mechanisms [40]. As of the end of last year, eleven bispecific inhibitors have been approved for the treatment of cancer. These inhibitors are available in different formats, address different targets, and mediate their anticancer functions via different molecular mechanisms. Data available from ClinicalTrials.gov, Cortellis, and The Antibody Society revealed more than 300 clinical trials involving 200 different bispecific molecules, with approximately 75% used to treat solid tumors and 25% to treat hematological malignancies [40].

Increasing evidence has demonstrated that disease-associated phenotypes are frequently triggered by more than one signaling pathway. This observation explains the main limitation of natural monospecific antibodies. It has been suggested that diseased cells overcome growth inhibition or the induction of cytotoxicity through a single target or pathway by using a compensatory signaling pathway [40]. This limitation is less likely with the use of bsAbs, which are capable of simultaneously modulating different disease-associated signaling receptors and/or pathways. For example, amivantamab (JNJ-61186372) targets the epidermal growth factor receptor (EGFR) also known as HER) and the hepatocyte growth factor receptor (MET) [81]. Both receptors trigger the proliferation of non-small cell lung cancer (NSCLC) and, consequently, block NSCLC’s growth more effectively than blocking just one pathway. Amivantamab is approved to treat a subtype of NSCLC that carries EGFR insertion mutations in Exon 20 [81]. The limited number of publications on the present and future role of bispecific antibodies in immune therapy carries a note of optimism and hope to many cancer patients who have experienced disillusion with the therapeutic results of monospecific antibodies. This optimism can be justified by considering two characteristics of bsAbs. First, it is accepted that most forms of cancer are multifactorial, and their initiation and progression are linked to different mechanisms and different signaling pathways. These characteristics are partially met in the design of bispecific antibodies. Furthermore, these inhibitors are available in different formats, address different targets, and mediate their anticancer function via different molecular mechanisms. Second, in recent years, the bsAbs field has been transformed from basic research to applied clinical applications, including drug development, which opens the door for new and exciting possibilities for immune therapy. A more realistic evaluation of the role of bsAbs in immune therapy has to await the outcome of many ongoing clinical trials, meta-studies, and large-scale statistical surveys on the extent of success or failure of this highly promising therapeutic approach.

#### 2.4.3. MS-Based Investigation of Immune Checkpoints Is Still below Its Real Potential

Over the last 10 years, immunotherapy has assumed a prominent role in the cure and management of various forms of cancer. This therapeutic approach is based on the blockade of receptors called immune checkpoints (ICs). Research efforts over the same period have established that the key players behind the mechanism(s) of therapy are glycoproteins, their encoding genes, and the associated signaling pathways. The title of this section refers to the use of mass spectrometry in the characterization of immune checkpoints. These proteins include PD-1, PD-L1, CTLA-4, BTLA, and CD160. We have already pointed out in this review that MS-based analyses can offer much-needed information on proteins’ expression, structure, PTMs, conformation dynamics in solution, and protein–protein and protein–ligand interactions. These information are considered to be central to a better understanding of the biology of the disease; without such an understanding, it would be difficult to design specific checkpoint inhibitors. The current literature indicates that the use of this powerful technique to investigate ICs and their ligands is still very limited. Such limited application can be partially attributed to the following observation. Most checkpoints are transmembrane glycoproteins, which render them insoluble in water. This insolubility renders their purification and crystallization in preparation for their analysis highly challenging. In recent years, mass spectrometry (MS) has assumed a significant role in the characterization of membrane proteins. This emerging role can be attributed to a number of reasons, including the availability of MS-compatible detergents that are able to efficiently solubilize membrane proteins within the investigated samples, maintaining their stability in solution. These detergents have the advantage of being easily removed after the transfer of the precursor ions from solution into the gas phase in the mass spectrometer without significantly impacting on the structure or stoichiometry of the investigated proteins [82,83].

MS-based proteomics is contributing to research efforts to understand the role of ICs in immune therapy. However, such a contribution remains insufficient and does not include key members of these proteins. Mass spectrometry-based investigation of BTLA is a representative example of the current status. BTLA is a transmembrane glycoprotein composed of 289 amino acids. It shares structural similarities with other immune checkpoint proteins such as PD-1 and CTLA-4. The latter two checkpoints and some of their PTMs have been investigated using MS-based methods; however, the search for similar investigations of BTLA has proved negative. The absence of such investigations is rather surprising. The interaction of this protein with the herpes virus entry mediator (HVEM) plays an essential role in negatively regulating immune responses, thereby preserving immune homeostasis. Future MS-based investigation of BTLA and its complexation with HVEM will no doubt enrich information derived from other sources on immune therapy and the search for more specific checkpoint inhibitors. 

## 3. Conclusions

The binding of T cell immune checkpoint proteins to their ligands allows immune evasion by tumors. To enhance the immune response against such evasion, a number of therapeutic antibodies that bind to these proteins or their ligands have been developed. The initial therapeutic impact of these inhibitors has been very positive; however, this initial success was rapidly eroded due to associated severe side effects, the fact that they only benefited a small subset of patients, and their high costs. Currently, there are two areas within the field of immune therapy that are attracting intense research activities: (i) the identification of predictive biomarkers capable of reliably discriminating between patients who respond to antibody therapy and those who do not, and (ii) the development and testing of a new class of antibodies, which can enable novel mechanisms of action that cannot be achieved by the existing IgG-based antibodies. A well-recognized example of such development is bispecific antibodies. These molecules are available in different formats, engage different targets, and mediate their anticancer function via different molecular mechanisms. The complexity of both arguments calls for a combination of various techniques and different technologies. These techniques include genomics, epigenomics, transcriptomics, molecular modeling, cutting-edge bioinformatics, and MS-based proteomics. These research efforts have to be supported by timely and well-designed clinical trials, involving the highest number of patients suffering different forms of cancer. As pointed out earlier in this text, the interaction between proteins and antigens associated with these checkpoints and the immune system is an important component in the mechanism of action of the existing checkpoint inhibitors. This means that to understand the mechanism(s) of resistance to these inhibitors, detailed information on the structure, conformation, and interactions of these proteins with other proteins is highly relevant. MS-based proteomics is well known for its capability to deliver such information. That said, the current literature has indicated that the contribution of this powerful technique to the characterization of the proteins and antigens involved in interaction of immune checkpoints and the immune system remains very limited. Both earlier as well as more recent works on the role of immune checkpoints in immunotherapy have underlined a number of future challenges. These challenges include a better understanding of the resistance mechanisms to the blockade of immune checkpoints, the identification of more efficient inhibitors and extending their therapeutic benefits to a wider range of cancer patients, better management of immune-related adverse side effects, and, more urgently, the identification of predictive biomarkers which would help treating oncologists in the identification of patients who are likely to respond positively to the immune therapies. The latter challenge is closely linked to the identification and validation of predictive biomarkers in response to ICI therapy. Addressing these challenges will require the combined efforts of basic researchers and clinicians.

## Figures and Tables

**Figure 1 ijms-25-09276-f001:**
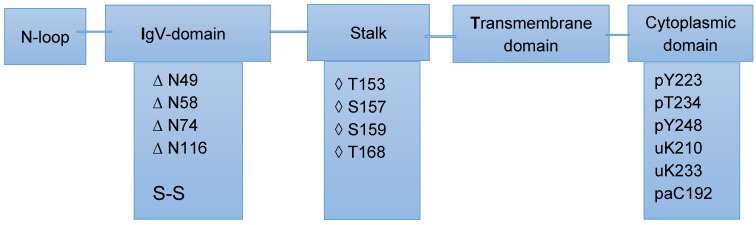
PD-1 domains and associated post-translational modifications. N- and O-glycosylation: are indicated with the symbols ∆ and ◊, respectively. The cytoplasmic domain experiences multiple modifications, such as phosphorylation(p), ubiquitarian (u), and palmitoylation (pa). N-glycosylation and O-glycosylation are observed at the IgV-domain and the stalk, respectively. An S-S bridge modification between C54 and C123 is also observed in the IgV domain. Based on [49,53].

**Figure 2 ijms-25-09276-f002:**
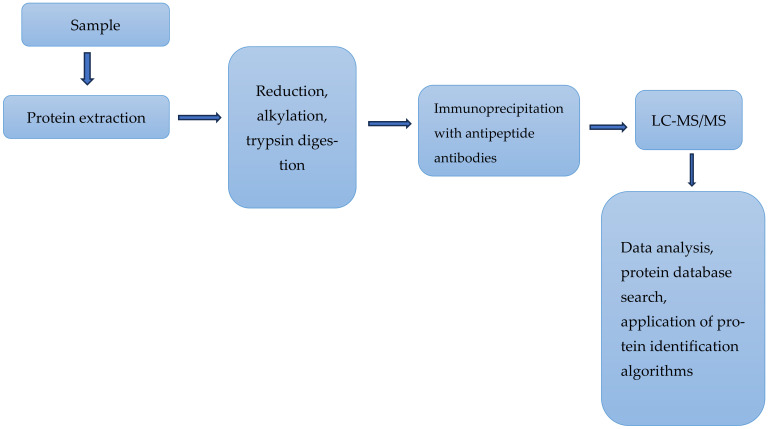
Immunoprecipitation of peptides derived from a biological sample followed by LC-MS/MS analyses for the quantitative measurement of CTLA-4 in human T cells. Based on [63].

**Figure 3 ijms-25-09276-f003:**
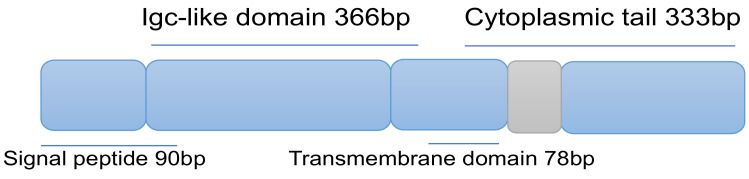
Schematic representation of BTLA’s structure, showing the three domains and the associated number of base pairs (bp). The structure is based on [69].

**Table 1 ijms-25-09276-t001:** Mass spectrometry-based analyses for the identification of potential predictive biomarkers in response to immune checkpoint inhibitors (ICIs).

Reference	Method of Analysis	(a) Tumor, (b) Therapy	Potential Biomarkers in Response to ICI Therapy
[23]	1. Trapped ion mobility spectrometry coupled with tandem mass spectrometry (MS/MS). 2. RNA-sequence analysis. Both analyses in 1 and 2 were supported with machine learning algorithms.	(a) Non-small cell lung cancer (NSCLC). (b) antiPD-1/PD-L1 therapy	Gene expression profile: *MOXD1, PHAF1*, *KRT7*, *ANKRD30A*, *TMEM184A*, *KIR3DL1*, and *KCNK4* According to the authors, the above profile predicted a durable response to anti-PD-1/PD-L1
[35]	LC-MS/MS).	(a) Non-small cell lung cancer (b) Anti PD-1 therapy	A metabolite panel consisting of hypoxanthine and histidine, identified in serum samples.
[36]	LC-MS/MS	(a) Melanoma (b) Tumor-infiltrating lymphocyte (TIL)-based therapy and anti-PD-1 therapy	Lipid and ketone body metabolism proteins in cancer cells
[37]	LC-MS/MS	(a) Gastric cancer (b) Anti PD-1 therapy	A high abundance of activated CD8 T cells. Using machine learning, a set of 10 proteins was identified as potential biomarkers: COL15A1, SAMHD1, DHX15, PTDSS1, CFI, ORM2, VWF, APOA1, EMC2, and COL6A2
[38]	High-resolution isoelectric focusing liquid chromatography–tandem mass spectrometry (HiRIEF LC-MS/MS)	(a) Metastatic cutaneous melanoma. (b) Anti-PD-1 therapy	1. An increase in circulating PD-1 was observed during anti-PD-1 treatment. 2. Anti-PD-1 responders had an increase in plasma proteins involved in the T cell response, neutrophil degranulation, inflammation, cell adhesion, and immune suppression. 3. An association between plasma proteins and progression-free survival (PFS). The proteins included interleukin 6; interleukin 10; proline-rich acidic protein 1; desmocollin 3; C-C motif chemokine ligands 2, 3 and 4; and vascular endothelial growth factor A

**Table 2 ijms-25-09276-t002:** Approved bispecific antibodies for cancer therapy and other diseases. Hemlibra and Vabysmo are used for non-oncological indications. The information in Table 2 are based on [40] and on the guidance reported in the FDA official website.

Trade Name and Year of Approval	Indications	Approved in
Removab (2009)	Ovarian intraperitoneal ascites	Withdrawn in 2017.
Blincyto (2014)	To treat Philadelphia chromosome-negative relapsed or refractory B cell precursor acute lymphoblastic leukemia	USA, EU, Japan
Rybrevant (2021)	To treat locally advanced or metastatic non-small cell lung cancer with certain mutations	USA, EU
KImmtrak (2022)	To treat a form of unresectable or metastatic uveal melanoma	USA, EU
Lunsumio (2022)	To treat relapsed or refractory follicular lymphoma	USA, EU
Kaltanni (2022)	Hepatocellular carcinoma	China
Tecvayli (2022)	To treat relapsed or refractory multiple myeloma	USA, EU
Columvi (2023)	To treat relapsed or refractory diffuse large B cell lymphoma or large B cell lymphoma	USA, EU
Epkinly (2023)	To treat relapsed or refractory diffuse large B cell lymphoma	USA, EU, Japan
Talvey (2023)	Relapsed/refractory multiple myeloma	USA
Elrexfio (2023)	Relapsed/refractory multiple myeloma	USA, EU
Hemlibra (2017)	To prevent or reduce the frequency of bleeding episodes in hemophilia A with factor VIII inhibitors	USA, EU, Japan
Vabysmo (2022)	To treat neovascular (wet) age-related macular degenerated and diabetic macular edema	USA, EU, Japan

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
