# Peer review of "Proteomic Investigation of Immune Checkpoints and Some of Their Inhibitors"

_ijms, 2024, doi:10.3390/ijms25179276_

Round 1

Reviewer 1 Report

Comments and Suggestions for Authors

In Agostini et al., the authors presented a comprehensive summary regarding the application of proteomics immune checkpoints blockade (ICB). The topic is very interesting as the contribution of proteomics in immune therapy indeed is insufficient. However, the paper sometimes lacks smooth transitions between sections, making it difficult to follow the progression of ideas. For instance, in the last paragraph of the introduction, the authors jumped to PD-1 elaboration. The reviewer understood that the author would like to exemplify the point, however, overstatements distracted the audience from initial purpose.

Using proteomics to predict biomarkers in cancer treatment, especially in treating metastasis tumors are very informative. It would be useful to delve deeper into potential identified genes or protein, or mutations (in addition to SAP, POLE/POLD1) to hint researchers on novel anti-tumor targets. A summary table or figure would be useful. In addition, adding meta-analysis on proteomic analysis with responsive versus non-responsive patients treated with ICB would be more informative.

In the session of 'drug resistance to immune therapies', the reviewer did see how proteomics contribute or involved, which supposed to be key discussion points in the paper. The authors need to provide insights on how proteomics to overcome the current hurdle instead of stating the current challenges and limitations. It would be helpful to include more specific examples of how MS has been successfully applied in cancer research to provide concrete evidence of its utility.

Besides, there are a few of typos or errors across the context that need to be fixed as listed below:

Line 18, additional '.' in the middle of sentence.

Line 23, 'Immune-checkpoints' should be 'immune-checkpoints'

Line 25, remove red color of ','.

Line 62, comma found at the end of sentence.

Line 142, period in the middle

Line 319, citation format

Line 508, wrong format

Comments on the Quality of English Language

There are lot of typos and errors need to be fixed. 

Author Response

All additions/changes within the revised text are in blue color.

1.Improving sections transition: We do agree with the referee regarding the wrong location of the paragraph PD-1 elaboration. This paragraph (lines 76-91 in old text) has been moved to section 2.2 (lines 245-256); a section dedicated to PD-L1/PD-1. This modification necessitated the addition of Lines 75-84 in the introduction.

  1. Using proteomics to predict biomarkers: To strengthen this section we have added table 1, inserted and commented a second meta study (lines 204-2017 in the revised text). The works cited in table1 date between 2021-2024.
  2. Session on drugs resistance: Lines 392-400 have been added.
  3. Errors listed by referee1 have been corrected in the new text.

Line 18, additional '.' in the middle of sentence.

Line 23, 'Immune-checkpoints' should be 'immune-checkpoints'

Line 25, remove red color of ','.

Line 62, comma found at the end of sentence.

Line 142, period in the middle

Line 319, citation format

Line 508, wrong format

The authors want to thank both referees for their constructive suggestions.

Reviewer 2 Report

Comments and Suggestions for Authors

This article titled as “Proteomic Investigation of Immune Checkpoints and some of Their Inhibitors” tries to discuss some of checkpoints, their inhibitors and some works in which mass spectrometry based proteomic analyses. Authors spent lots of effort and content to explain the meaning and identification of Proteomic Investigation of immune checkpoints.  Authors mentioned that “MS-based analyses can offer much needed information on protein expression, structures…..”. However,  The manuscript does not provide any practical content, such as the actual significance of clinical applications or how they can be applied to tumor diagnosis or efficacy prediction. The authors should provide a detailed summary of relevant applications in different types of tumors, including specific clinical data, future research directions, clinical applications, and potential challenges. I recommend major revisions. The authors should address the comments above to enhance the clarity, coherence, and impact of their work.

Author Response

All additions/changes within the revised text are in blue color.

Referee 2.

Referee 2 asks for more relevant applications of MS-based proteomics in the investigation of different tumors with emphasis on diagnosis and efficacy of predictive biomarkers.

(a)The addition of table 1 in the revised version is part of the answer to the above suggestion. The works cited in this table summarizes recent MS-based works for the identification of predictive biomarkers associated with various forms of cancer, including non-small cell lung cancer (NSCLC); esophageal squamous cell carcinoma and Melanoma.

(b) The meta study, lines 230-243 (Ref. 33) reported the results of 100 peer-reviewed studies with data from 18,792 patients.  Suffering from various forms of cancer, including, non-small cell lung cancer, melanoma, urothelial cancer, head and neck cancer and colorectal cancer. The cited reviews demonstrated that tumor mutational burden (TMB) and PD-L1IHC as predictive biomarkers adequately captured responders and non-responders across all investigated cancer types.

(c) To underline the contribution of MS-based proteomics to biomarkers discovery and validation we have added lines 177-201 and references 23-27.

(d) potential challenges. Lines 583--593 have been added to conclusions.

Round 2

Reviewer 1 Report

Comments and Suggestions for Authors

The authors have addressed the all the points and it is good for publication. However, there are some formats inconsistency (font, size) needs to be fixed. For example, line 23, line75-96. 

Author Response

Referee 1 pointed out format’s inconsistency.

Correction to some of these insistencies have been corrected in various places in the text.

Reviewer 2 Report

Comments and Suggestions for Authors

The manuscript titled as "Proteomic Investigation of Immune Checkpoints and Some of Their Inhibitors" provides a comprehensive overview of the role of immune checkpoints in cancer therapy, highlighting the advancements and challenges associated with the use of ICIs. The use of mass spectrometry (MS) in proteomics to understand immune checkpoints is thoroughly discussed. Several areas still require clarification, additional information, and minor corrections.

In the subsection "Predictive Biomarkers in Response to Immune Checkpoint Inhibitors" (lines 125-244), more emphasis is recommended to be placed on the clinical implications of these biomarkers. How can they be applied in clinical practice to improve patient outcomes?

The section on bispecific antibodies needs a comparison with conventional antibodies and a discussion on their potential limitations.

The conclusion section should expanded to include more specific recommendations for future research. What are the key gaps in the current knowledge, and how should future studies address them?

Line 19: "and disease progress" should be "and disease progression."

Line 27: "Numerous studies have demonstrated" – consider specifying the number or nature of these studies for clarity.

Line 39: "gave a fresh hope" should be "gave fresh hope."

Author Response

2nd Referee round 2

In the subsection "Predictive Biomarkers. To address this comment, we have added lines 200-207 in blue.

The section on bispecific antibodies. To address this comment, we have added lines 407-414 and reference 78 (in blue).

The conclusion section. To address this comment, we have added lines 504-511(in blue).

 Line 19. Corrected.

 Line 27: "Numerous studies have demonstrated. Corrected and modified lines 27-30 (in blue).

 Line 39: corrected